# Efficacy of Oral Rabies Vaccine Baits Containing SPBN GASGAS in Domestic Dogs According to International Standards

**DOI:** 10.3390/vaccines11020307

**Published:** 2023-01-30

**Authors:** Katharina Bobe, Steffen Ortmann, Christian Kaiser, David Perez-Bravo, Jörn Gethmann, Jeannette Kliemt, Sophia Körner, Tobias Theuß, Thomas Lindner, Conrad Freuling, Thomas Müller, Ad Vos

**Affiliations:** 1Ceva Innovation Center, Am Pharmapark, (D.P.-B. formerly: Institute of Epidemiology, Friedrich-Loeffler-Institute), 06861 Dessau-Rosslau, Germany; 2TEW Servicegesellschaft GmbH, Am Pharmapark, (C.K. formerly: Ceva Innovation Center), 06861 Dessau-Rosslau, Germany; 3Institute of Epidemiology, Friedrich-Loeffler-Institute, 17493 Greifswald-Insel Riems, Germany; 4Institute of Molecular Virology and Cell Biology, Friedrich-Loeffler-Institute, 17493 Greifswald-Insel Riems, Germany

**Keywords:** dog, rabies, oral vaccination, challenge, efficacy study, SPBN GASGAS

## Abstract

(1) Background: The oral vaccination of free-roaming dogs against rabies has been developed as a promising complementary tool for mass dog vaccination. However, no oral rabies vaccine has provided efficacy data in dogs according to international standards. (2) Methods: To test the immunogenicity and efficacy of the third-generation oral rabies virus vaccine strain, SPBN GASGAS, in domestic dogs, dogs were offered an egg-flavoured bait containing 3.0 mL of the vaccine (10^7.5^ FFU/mL) or a placebo egg-flavoured bait. Subsequently, these 25 vaccinated and 10 control animals were challenged approximately 6 months later with a dog rabies virus isolate. Blood samples were collected at different time points postvaccination and examined by ELISA and RFFIT. (3) Results: All but 1 of the 25 vaccinated dogs survived the challenge infection; meanwhile, all 10 control dogs succumbed to rabies. The serology results showed that all 25 vaccinated dogs seroconverted in ELISA (>40% PB); meanwhile, only 13 of the 25 vaccinated dogs tested seropositive ≥ 0.5 IU/mL) in RFFIT. (4) Conclusions: The SPBN GASGAS rabies virus vaccine meets the efficacy requirements for live oral rabies vaccines as laid down by the European Pharmacopoeia and the WOAH Terrestrial Manual. SPBN GASGAS already fulfilled the safety requirements for oral rabies vaccines targeted at dogs. Hence, the egg-flavoured bait containing SPBN GASGAS is the first oral vaccine bait that complies with WOAH recommendations for the intended use of oral vaccination of free-roaming dogs against rabies.

## 1. Introduction

Oral rabies vaccination (ORV) is the method of choice to control the disease in specific wildlife reservoir species such as red fox (*Vulpes vulpes*), raccoon dog (*Nyctereutes procyonoides*), coyote (*Canis latrans*), golden jackal (*Canis aureus*) and raccoon (*Procyon lotor*) in many countries [1,2,3,4]. Several oral rabies vaccines have been developed, licensed, and are commercially available [5]. ORV has also been suggested for owned and ownerless free-roaming dogs that cannot be easily restrained for parenteral vaccination. Many laboratory and field studies were conducted and evaluated under guidance of the World Health Organization (WHO) in the 1990s [6,7,8,9,10,11]. However, these promising initiatives were abandoned, and only recently renewed interest and support for ORV of dogs was expressed by members of the rabies community [12,13,14,15,16,17]. Unfortunately, none of the available commercial products have obtained a license for using these products in dogs (*Canis lupus familiaris*). Off-label use of vaccines commonly occurs, including targeting species not stipulated in the Summary of Product Characteristics (SPC). However, for rabies with an almost 100% case fatality rate, very stringent standards for vaccines are established. The fact that none of the licensed oral rabies vaccines have listed the dog as target species renders it more difficult to use these products for the ORV of free-roaming dogs in most dog-rabies-endemic countries. Besides safety studies in target and nontarget species, it is essential that the vaccine should also be highly efficacious. It should induce a protective immune response after the consumption of a single vaccine bait. Part of the required testing is a study to show protection of vaccinated dogs via a challenge infection after a predetermined period. The regulatory authorities have provided detailed requirements concerning such an efficacy study. For example, the European Pharmacopoeia (Pharm. Eu.) requires a minimum number of 25 vaccinated and 10 control animals, a minimum period between vaccination and challenge infection (≥180 days), and a minimum percentage of vaccinated (92%) and control animals (90%) that survive and succumb to infection [18]. Small deviations do exist between the different sources. For example, in the Terrestrial Manual of the World Organization for Animal Health (WOAH), only 88% of the vaccinated dogs need to survive, and 80% of the control animals should succumb to rabies after the challenge infection [19]. So far, only immunogenicity studies or challenge studies in dogs with a limited number of animals and/or a shorter observation period have been conducted with different oral rabies vaccine baits [20,21,22,23,24]. In order to meet the requirements stipulated for such vaccine baits, a challenge study in dogs as required by the European Medicines Agency (EMA) and WOAH has been conducted with a third-generation oral rabies virus vaccine, SPBN GASGAS. This vaccine is registered in the European Union (EU) since December 2017 and used for the ORV of foxes and raccoon dogs in several EU member states [25]. It has also been intensively studied for the ORV of free-roaming dogs during experimental and field studies in different countries [26,27,28,29,30]. With the successful completion of the efficacy study presented here, this vaccine virus meets all required safety and efficacy standards, thus, hopefully eliminating the last hurdle for the integration of the ORV of dogs in areas with high proportions of dogs that are inaccessible for parenteral vaccination.

## 2. Materials and Methods

### 2.1. Vaccine Bait

The SPBN GASGAS vaccine suspension (10^7.5^ FFU/mL, 3.0 mL) was filled in flexible sachets made of a proprietary foil (“soft blister”) and embedded in egg-flavoured gelatine-based baits. The SPBN GASGAS vaccine construct is derived from a cDNA clone (SAD L16) of the vaccine strain SAD B19. Through site-directed mutagenesis, the residual pathogenicity observed with the parental strain in mice after i.c. inoculation has been abolished (amino acid exchange at amino acid position 333 of the glycoprotein gene). Additionally, all 3 nucleotides at amino acid position 194 of the glycoprotein gene were exchanged to circumvent, potentially, a partial reversion to virulence due to a compensatory mutation at this position. Finally, a second identically modified glycoprotein gene was inserted between the G- and L-gene [31,32,33].

### 2.2. Challenge Virus

For the challenge infection, a dog rabies isolate from Azerbaijan (FLI-Lab ID: 5989, 2004; 2nd Mouse NA- cell passage from 28 August 2012, 10^5.3^ MICLD_50_/mL) was used (GenBank: LN879480.1) [34,35]. This challenge virus had been tested during previous screening challenge studies in different animal models, including dogs (unpublished results). The MICLD_50_/mL virus stock was determined as described [36,37].

### 2.3. Animals

A total of 40 beagles (breed: Beagle HsdRcc: DOBE, source: Envigo Global Services Inc., Denver, PA, USA) were used. The animals were at least 8 weeks old upon arrival. Sex ratio was 1:1. All dogs were clinically healthy, and the animals were individually marked with a transponder chip for identification purposes. The regulatory requirements stipulate a minimum number of animals to be included in the challenge phase of the study. Therefore, 5 additional [surplus] animals (3 vaccinated and 2 controls) were included in case of unforeseen losses prior to challenge infection. However, in the final challenge phase, only 25 vaccinated and 10 control animals were infected.

### 2.4. Housing Conditions

Upon arrival, and until the challenge infection, all animals were located in an outdoor animal enclosure. The inside area per cage was 6–8 m^2^, and the outside area per cage was 17–55 m^2^. Environmental enrichment items like pull ropes, chewing toys, cloths and dog beds were provided in each cage. If necessary, the inside and outside area of each cage could be separated from each other. The animals were divided by sex in equal groups with a maximum of 5 animals per cage. For the time of vaccine administration, the animals were separated from the others for at least 2–4 h, avoiding possible horizontal transmission. At the time of the challenge, the animals were transferred from the outdoor enclosure to the units of the indoor experimental animal facility at Ceva Innovation Center, Dessau-Rosslau, Germany. The animals were kept in individual cages (ground area: 4 m^2^). As these units had a total housing capacity for 20 dogs, it was necessary to split the study into 2 parts.

The animals were fed twice a day (Vollmers’ Welpenkost, Gerhard Vollmer GmbH &Co. KG, Spenge, Germany). Water was offered ad libitum. All dogs were observed at least once daily for general health, feed intake and defecation by the staff. In the case of an abnormal clinical finding during the daily animal check, it was determined if the observation was due to a rabies infection and, if not, if the animal needed treatment. In case of severe clinical signs, dogs were humanely killed.

Treatment and sampling in general were done without anesthesia or sedation. Only for the challenge infection, and on single occasions for blood sampling, animals were sedated using Medetomidine (Domitor^®^, Vetoguinol, Ismaning, Germany) at a dosage of 0.1 mg/kg body weight i.m. during the first part. During the second part of the study, Tiletamin hydrochloride + Zolazepam hydrochloride (Zoletil 100 [50 mg/mL + 50 mg/mL], Virbac Arzneimittel GmbH, Bad Oldesloe, Germany) was used in a dosage of 7.5 mg/kg. Induction of euthanasia was done by Xylazine (2%, 2 mg/kg) and Ketamine (10%, 10 mg/kg) i.m. (Serumwerk Bernburg AG, Bernburg, Germany). For euthanasia, T61^®^ (Intervet GmbH, Unterschleissheim, Germany) was used in deep general anesthesia and was administered at a dosage of 0.3 mL/kg body weight.

At the end of the experiment, all dogs that survived the challenge were humanely killed, as described above, except for the 5 surplus animals that were not challenged. These animals were handed over to new owners through a specialized organization (Labor-Beagle-Hilfe e.V.) after termination of the study.

### 2.5. Vaccination Protocol

The dogs were allocated to two treatment groups: Group 1 (n = 28, 25 + 3 surplus animals) received the egg-flavoured vaccine bait, Group 2 (n = 12, 10 + 2 surplus animals) received an egg-flavoured placebo bait (sachet filled with water). The study was split into 2 parts (Phase 1 & 2: both 14 vaccinated and 6 control animals). The animals in phase 2 were vaccinated 4 months after the animals in phase 1. The study was blinded and only the vaccinator knew which animals were vaccinated or not. All of the other people involved, including study monitor and animal keepers, did not know which animal received which treatment. Moreover, all samples collected were tested blindly without knowledge on treatment.

### 2.6. Sampling Protocol

Blood samples were taken 14 days before vaccination (B0: −14 days postvaccination [dpv]) and at 10 (B1), 16/17 (B2), 31 (B3), 45 (B4), 59 (B5), 94 (B6), 129 (B7), 157 (B8), 185 dpv (B9) and at the study end post challenge (B10). The challenge infection was postponed for 14 days during the second study phase due to a technical problem with the wastewater inactivator. Hence, an additional blood sample was collected from the animals involved at 199 dpv (B9*). Blood samples B0 and B3 were tested directly after collection (ELISA). Consequently, it was known that all animals tested negative for rabies antibodies prior to vaccination and that, approximately 1 month postvaccination, 28 tested seropositive and 12 dogs tested seronegative. Based on these results, it was decided that all 5 surplus animals did not have to be included in the last part of the study and could be rehomed in case the other 35 animals survived the study period prior to challenge. The surplus animals were selected based on the last 3 digits of their identification number (chip); the 3 vaccinated and 2 control dogs with the highest 3-digit numbers were identified as surplus animals.

Blood samples (approximately 5–6 mL per sample) were taken from the large superficial veins of the extremities (e.g., *V. cephalica antebrachii*, *V. saphena*), generally without sedation using blood collection tubes (S-Monovette, Sarstedt, Nümbrecht, Germany). The final blood samples at the end of the study (just before euthanasia) were obtained under deep general anesthesia by intracardiac puncture. Blood samples were centrifuged at 3500 G for 15 min, and, subsequently, sera were stored between −20 °C and −35 °C until laboratory analysis for the presence of antibodies.

### 2.7. Challenge Protocol

The animals in phases 1 and 2 were challenged 185 and 199 dpv, respectively. In phase 1, 13 vaccinated and 5 control dogs were challenged, and, in the second phase, 12 vaccinated and 5 control animals were infected with the challenge virus. The diluted challenge virus administered had a titer of 10^3.6^ MICLD_50_/mL. One (1.0) ml of this material was administered intramuscularly, with 0.5 mL in the right and left M. masseter each. 

After the challenge virus inoculation, the dogs were monitored at least twice daily for clinical signs of rabies, and clinical scores were recorded, as described previously [38]. The animals were euthanised as soon as clinical score 2 or higher was reported.

Brain samples were taken for rabies diagnosis from each of the challenged animals that were euthanized after showing clinical signs and all remaining animals at the end of the 91-day period postinfection.

### 2.8. Diagnostic Assays

#### 2.8.1. Blocking Enzyme Linked Immunesorbent Assay (bELISA)

For the detection of rabies-specific binding antibodies, a commercially available bELISA (BioPro Rabies ELISA, Praha, Czech Republic) was used as described [39,40]. In brief, serum samples were incubated on microtiter plates coated with the rabies antigen. After removing the sera, all wells were incubated with a fixed amount of biotin-labeled rabies-specific antibodies. The bound antibodies were then incubated with peroxidase-conjugated streptavidin followed by chromophoric detection. As per manufacturers’ instructions, the test was validated for the detection of rabies-specific antibodies, with a percentage of blocking (PB) lower than 40% considered as negative and a PB equal or higher than 40% considered as positive.

#### 2.8.2. Rapid Fluorescence Focus Inhibition Tests (RFFIT)

Sera were tested for the presence of virus-neutralizing antibodies (VNA) in a modified Rapid Fluorescent Focus Inhibition Test (RFFIT), as described previously [41], using RABV (CVS-11) as the test virus and BHK21-BSR/5 (CCLV-RIE 0194/260) cells. The calibrated WHO international standard immunoglobulin (2nd human rabies immunoglobulin preparation, National Institute for Standards and Control, Potters Bar, Hertfordshire, UK) adjusted to 0.5 international units (IU) and a naive serum served as positive and negative controls, respectively. The VNA titres were calculated by fitting a sigmoidal function using nonlinear regression as implemented in R studio [42,43] and, subsequently, converted into concentrations expressed in IU/mL [44]. As a cut-off for seropositivity in the RFFIT a value of 0.5 IU/mL was used.

#### 2.8.3. Rabies Laboratory Diagnosis

Rabies was confirmed by the detection of the rabies antigen in brain samples using the fluorescent antibody test (FAT) [45]. The procedure involved that a piece of brain was smeared on a slide, air dried and fixed. The tissue slide was stained with a commercially available FITC anti-rabies hyperimmune serum (Monoclonal Anti-Rabies, FITC, SIFIN Diagnostics GmbH, Berlin, Germany) and examined under a fluorescence microscope. For questionable results, the brain sample was also investigated for the presence of RABV-specific RNA using the RT-qPCR [46] and Rabies Tissue Culture Infection Test (RTCIT) for virus isolation [47].

#### 2.8.4. Compliance with Ethical Standards

All applicable international, national and/or institutional guidelines for the care and use of animals were followed. Animal housing was conducted according to applicable guidelines. The study was approved by the competent authorities of the Land of Saxony-Anhalt, Germany (203.m-42502-2-1629 CEVA).

### 2.9. Statistical Analysis

For the calculation and graphics of (geometric) mean titres and 95% confidence intervals, GraphPad Prism 9.0 (GraphPad Software Inc., San Diego, CA, USA) was used.

## 3. Results

All dogs offered a vaccine bait accepted it within 10 min and seroconverted by 28 dpv and remained seropositive in ELISA during the entire study period. Meanwhile, all control animals remained seronegative during the observation period, and, only after the challenge infection, 6 out of 10 control animals seroconverted (ELISA) (Figure 1 and Table 1). Blood samples were also collected from the surplus animals, but, as these animals were not included in the final challenge study, the results have not been incorporated in the following analysis. However, all three surplus dogs offered a bait seroconverted, and both control surplus dogs remained seronegative (ELISA). In vaccinated dogs, an increase in PB was observed after the challenge; moreover, eight of the ten control animals tested seropositive (>40%PB) on the day they were euthanized postinfection.

The results were less clear using the RFFIT (Table 2). The highest seroconversion rate was obtained 45 dpv; 13 of the 25 orally vaccinated animals (52%) had VNA levels > 0.5 IU/mL. Seven vaccinated dogs never reached VNA-levels > 0.5 IU/mL during the entire period between vaccination and challenge infection, and two of those never exceeded 0.25 IU/mL. Two control animals had VNA levels above the threshold of positivity on a single sampling occasion: B4—1.74 IU/mL and B6—0.56 IU/mL. As shown previously, both sera tested negative for rabies-specific antibodies in ELISA.

Just as with the ELISA, also a booster effect as a result of the challenge infection was noticed in 76% of the vaccinated dogs. Seven of the ten control animals also had VNA-levels > 0.5 IU/mL at the day of death.

All vaccinated dogs survived the challenge infection, except one dog that was euthanized 12 days postinfection based on the severity of clinical neurological symptoms observed in the animal. The animal tested negative for the presence of the RABV antigen in the brain using FAT. Subsequently, the brain was also tested in RT-qPCR and RTCIT. No virus was isolated in the cell culture (RTCIT), but it tested positive for the presence of RABV RNA in RT-qPCR. Furthermore, sequence analysis showed 100% identity with the challenge virus used. The animal developed a consistent, detectable immune response throughout the observation period in ELISA, but, in the RFFIT, the VNA-levels prior to the challenge never exceeded 0.5 IU/mL. The other six vaccinated dogs that were considered seronegative (<0.5 IU/mL) in the RFFIT all survived the challenge infection. All 10 control dogs were euthanized between 10 and 14 days postinfection after showing clinical signs and tested positive for the presence of the RABV antigen in the brain using the FAT.

## 4. Discussion

Even though serum antibody titres against RABV are considered a surrogate for predicting protective immunity after rabies vaccination [48], an efficacy study including a challenge infection in both vaccinated and control animals is required for regulatory purposes. Our efficacy study with SPBN GASGAS met the requirements of Pharm EU and WOAH [18,19], as 96% (24/25) of the vaccinated dogs survived, and 100% (10/10) of the control animals succumbed from the challenge infection. These results corroborate the principal utility of this vaccine construct, as previous efficacy studies showed that SPBN GASGAS protected red foxes and raccoon dogs, the two reservoir species in Europe, against a severe challenge [38]. Concerning the onset of immunity, in this study, 88% (22/25) and 100% (25/25) of all vaccinated dogs seroconverted (ELISA) approximately 2 weeks and 1 month postvaccination, respectively. The same was observed during a study in Thailand where 100% seroconversion rate in dogs orally vaccinated with SPBN GASGAS was also not obtained before 28 dpv, whereas all parenterally vaccinated animals had seroconverted already 14 dpv [30]. Hence, it seems that the onset of a measurable immune response in orally vaccinated animals is slower than in parenterally vaccinated animals. Our study confirmed a previous comparative study that the results of the commercially available ELISA kit used are better suited as a correlate of protective immunity than the RFFIT [41]. All vaccinated and control animals tested seropositive and seronegative, respectively, in ELISA (Table 1). In contrast, the RFFIT results were less pronounced (Table 2) and even produced false positives in individual sera (n = 2) from unvaccinated animals while being negative in ELISA for rabies-specific binding antibodies [49,50,51]. There was also a near-perfect match with the outcome of the challenge study; seropositive animals survived and seronegative ones succumbed to rabies, with the exception of the one vaccinated animal that showed clinical signs. As for the RFFIT, six vaccinated dogs survived the challenge without developing VNA-levels > 0.5 IU/mL during any time point between vaccination and the challenge. Similar observations have been made in a dog challenge study using another oral rabies vaccine, SAG2 [21]. Based on the results of a long-term immunogenicity study conducted in Thailand, it can be assumed that the duration of immunity (DOI) for dogs after the consumption of a single bait will last at least 12 months [30]. This is supported by an additional challenge study with the same construct in foxes, where the animals were protected against a challenge infection 12 months postvaccination [52]. An immunogenicity study in foxes using another oral rabies virus vaccine, an ERA-strain, showed even partial protection against a rabies challenge infection for up to 7 years [53]. Hence, it is likely that oral vaccination with SPBN GASGAS will induce lifelong protection against rabies infection in most dogs, considering the relatively low life expectancy of free-roaming dogs. Various immunogenicity studies with SPBN GASGAS in dogs have been conducted in local dogs in different settings, whereby the conditions of how the bait offering was conducted seemed to impact the obtained seroconversion rate [26,28,30]. As this efficacy trial demonstrated, in addition to the similar results to these studies, there is little to no value in additional local serology, especially when it involves a challenge infection. From an animal welfare perspective, unnecessary testing involving the killing of dogs should be avoided. Furthermore, appropriate housing conditions according to WOAH standards [54] are not available in most dog-rabies-endemic countries. Instead, working towards the concept of vaccine regulatory convergence among WOAH member countries should be given the highest priority [12].

Besides efficacy and immunogenicity, another essential parameter for oral vaccines, particularly for dogs, is safety. Previously, it was already shown that SPBN GASGAS meets the minimum WOAH safety requirements for oral rabies vaccines [19]. An overdose and dissemination study including viral shedding provided evidence for the safety of the vaccine in dogs even at high doses. The vaccine virus did not disseminate beyond the site of entry and was not actively shed in saliva and faeces [55,56]. Furthermore, the vaccine construct was also shown to be safe for different nontarget species, including cats and rodents [55]. As required, there was no horizontal transmission in rodents when vaccinated and naïve control animals were placed in the same cage [55]. Moreover, the innocuity of the vaccine in immune-compromised hosts (SCIDand nude mice) after different routes of administration was examined [57]. Finally, an assessment to evaluate the safety risk for humans and an estimation of the likelihood that humans will come in contact with the vaccine virus, as required by WOAH, was conducted [57]. Furthermore, full genome sequencing and the genetic stability of the vaccine virus after serial passaging, in vivo and in vitro, was disclosed [58].

During former challenges and field studies with the first-generation oral rabies vaccine, SAD B19, a rigid PVC container with an aluminum cover foil was incorporated in a segment of boiled intestine [20,59,60]. Initially, the same intestine bait and PVC-container was used for SPBN GASGAS. The intestine bait was shown to be extremely attractive for dogs [26,61,62]. There were some safety concerns with the PVC-container used, as it could cause internal injuries and/or obstructions when swallowed by dogs. Furthermore, the preparation of the boiled intestine bait was deemed inconvenient and added a potential risk of a breach in the cold chain [27]. Hence, both vaccine container and bait were replaced by an egg-flavoured bait containing a soft blister. Bait acceptance studies showed that the egg-flavoured bait with soft blister was not only very attractive to the dogs but was also capable of timely release of the vaccine in the oral cavity during bait consumption [28,29,63]. These results were corroborated by this efficacy study, where all animals accepted and ingested the egg-flavoured bait readily and, subsequently, perforated the soft blister.

## 5. Conclusions

The SPBN GASGAS vaccine bait is the first oral rabies vaccine bait that complied with all safety and efficacy requirements of oral rabies vaccine baits for free-roaming dogs, as stipulated by the WOAH. Not only is the highly attenuated third-generation SPBN GASGAS vaccine construct safe, it is also highly efficacious when delivered in the selected soft blister incorporated in a very attractive egg-flavoured bait. Hence, the ORV of dogs with the SPBN GASGAS vaccine bait can be added safely and routinely to the set of tools used to reach a better vaccination coverage and the ambitious goal of the tripartite (WHO/WOAH/FAO) of eliminating dog-mediated human rabies by 2030.

## Figures and Tables

**Figure 1 vaccines-11-00307-f001:**
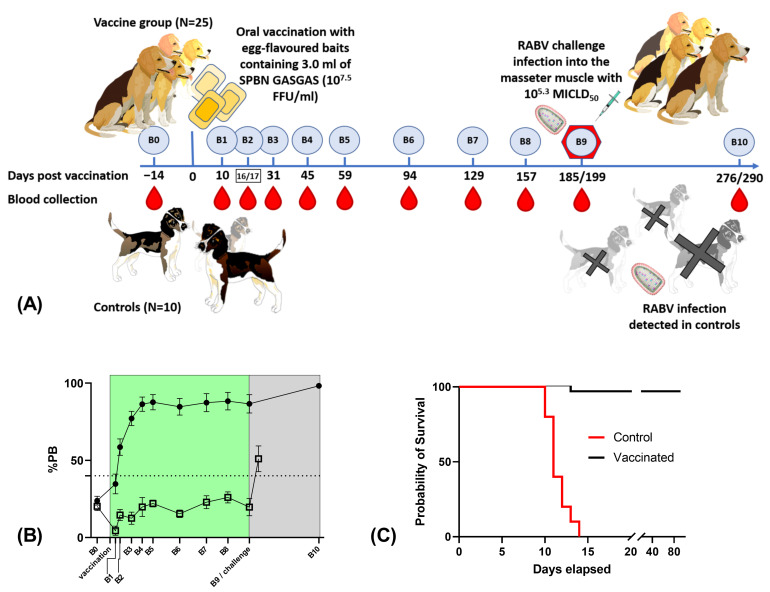
(**A**) Study protocol including vaccination, challenge infection and blood sampling; (**B**) The average percentage of blocking (%PB) and 95% confidence interval in vaccinated (solid circles) and control dogs (open squares) at the selected sampling points as determined by ELISA. The blood sample B10 was collected 11–91 days post challenge infection, depending on the day of euthanasia; (**C**) Survival probabilities of the two treatment groups.

**Table 1 vaccines-11-00307-t001:** Seroconversion rate (n/N) in vaccinated (V) and control (C) dogs at selected days postvaccination (dpv) based on the ELISA results; n—number of dogs seroconverted (≥40%PB), N—total number of dogs in treatment group.

	B0	B1	B2	B3	B4	B5	B6	B7	B8	B9	B10
dpv	−14	10	16/17	31	45	59	94	129	157	185	276/290 *
V	0/25	9/25	22/25	25/25	25/25	25/25	25/25	25/25	25/25	25/25	25/25
C	0/10	0/10	0/10	0/10	0/10	0/10	0/10	0/10	0/10	0/10	6/10

* B10 was collected 11–91 days post challenge infection, depending on the day of euthanasia, and, in the second phase, there was a 14-day delay in administration of challenge virus. Hence, B10 was collected 290 dpv instead of 276 dpv.

**Table 2 vaccines-11-00307-t002:** Seroconversion rate (n/N), Geometric Mean Titre (GMT—IU/mL), maximum and minimum level VNA in vaccinated (V) and control (C) dogs at selected days post vaccination (dpv) based on the RFFIT results; n—number of dogs seroconverted (>0.5 IU/mL), N—total number of dogs in treatment group.

	B0	B1	B2	B3	B4	B5	B6	B7	B8	B9	B10
dpv	−14	10	16/17	31	45	59	94	129	157	185	276/290 *
V	0/25	7/25	11/25	10/25	13/25	12/25	11/25	11/25	11/25	9/25	19/25
GMT	0.20	0.69	1.32	0.60	0.84	0.59	0.50	0.49	0.55	0.47	1.66
min	0.06	0.13	0.08	0.25	0.11	0.04	0.04	0.06	0.02	0.07	0.25
max	0.47	63.51	86.37	21.97	8.75	7.99	4.02	11.37	5.54	3.34	17.40
C	0/10	0/10	0/10	0/10	1/10	0/10	1/10	0/10	0/10	0/10	7/10

*—B10 was collected 11–91 days post challenge infection, depending on the day of euthanasia and in the second phase there was a 14-day delay in administration of challenge virus. Hence, B10 was collected 290 dpv instead of 276 dpv.

## Data Availability

The datasets used and/or analysed during the current study are available from the corresponding author on reasonable request.

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
