# Peer review of "Efficacy of Oral Rabies Vaccine Baits Containing SPBN GASGAS in Domestic Dogs According to International Standards"

_vaccines, 2023, doi:10.3390/vaccines11020307_

Round 1

Reviewer 1 Report

Congratulations to the authors for a well thought out and relevant study. The manuscript is well drafted. The purpose, methods and results are clearly presented. However, there are a few edits I would suggest, to improve comprehension.

Author Response

Please see below our reply to every single comment made by reveiwer 1:

Congratulations to the authors for a well thought out and relevant study. The manuscript is well drafted. The purpose, methods and results are clearly presented. However, there are a few edits I would suggest, to improve comprehension.

1) l22-24: Information on surplus animals in the abstract is confusing. It isn’t highly relevant information for the reader at this point. Please edit to convey information of the animals used in the experiment only.
Reply: we agree with the reviewer that the ‘story’ of the surplus animals is irrelevant for the abstract and we thus adapted the text accordingly. However, we would prefer to include the information in the main text as it underscores some of the additional ‘challenges’ associated with such a study.

2) l27-28: Please mention what the serology results indicated on immunogenicity of the vaccine, not the fact that ELISA is a better suited test than RFFIT.
Reply: we changed the sentence as requested

3) l37: Remove the word ‘nowadays’
Reply: deleted

4) l46: Reference 17 is not a valid reference. Please remove.
Reply: this reference has been replaced by a more relevant one.

5) l61: Mention the number of animals in brackets with %; maybe like this: (92%, 23/25) and (90%, 9/10)
Reply: Unfortunately, this is not possible as you may include more animals than 25 (vaccinated) or 10 (control); it is the percentage that counts not the absolute number (only the lower limit)

6) l64: Similarly; (88%, 22/25) and (80%, 8/10)
Reply: see previous reply

7) l78: This paragraph should be titled ‘Vaccine’. Please include a separate paragraph under a new subheading ‘bait’ or ‘vaccine bait’ describing the bait in brief. This information shall also help the reader in the discussion on bait and sachet, in last paragraph of discussion section.
Reply: we understand why the reviewer suggests this but again we try to align this publication with the report submitted to the regulatory authorities and the product to be tested is the final product and that is the vaccine bait. With other words the study concerns the efficacy of the vaccine bait, not the efficacy of the vaccine. Hence, we would prefer to stick to ‘vaccine bait’

8) l146: Improve sentence “Blood samples were taken prior to study start” to “ Blood samples were taken before the commencement of the study”. Was this blood sample collected 14 days before vaccination? This information is not clear.
Reply: we adapted the sentence so hopefully it is clear now.

9) l154: “-“ with the word negative is a typo. The authors could rework a few sentences in the manuscript to reduce the use of ‘respectively’.
reply: corrected and the sentence has been adapted to get rid of the word ‘respectively’

10) l162: Please add detail in bracket as shown for better comprehension. The final blood samples (just before euthanasia) at the end of the study were obtained under deep general anaesthesia by intracardiac puncture.
Reply: inserted

11) l169: Use the word ‘administered’ instead of ‘applied’
Reply: corrected

12) l200: Replace ‘Hereby’ with ‘The procedure involved’
Reply: accepted and changed

13) l203-206: Can be modified to “For questionable results, the brain sample was also investigated for the presence of RABV specific RNA by RT-qPCR [46] and Rabies Tissue Culture Infection Test (RTCIT) for virus isolation”
Reply: accepted and changed accordingly

14) l262: Please provide more information and references on sequencing technique
Reply: The reference for the sequencing technique [46] was given in the M&M section. We purposely did not go into too much detail concerning this particular animal as it would distract from the key message. We are afraid that this could turn into a discussion on the validity of the diagnostic assays and this is basically irrelevant here and this animal does not impact the main conclusion of this study. Therefore, we would prefer not to give more details here.   

15) l274: Mention the number of animals in brackets; (96%, 24/25) and (100%, 10/10)
Reply: changed accordingly

16) l278: Mention the number of animals in brackets
Reply: changed accordingly

17) l304: Edit: Hence, considering the high population turnover of the free-roaming dog population ‘leading to’ relatively low life expectancy, it is likely that oral vaccination with SPBN GASGAS will induce lifelong protection against rabies infection in most dogs.
Reply: Thanks for pointing this out, we changed the sentence and only include the low life expectancy as the population turnover has more to do with birth and death. In the context of this remark the low life expectancy is only of concern.

18) l310: Include ‘unnecessary testing/experimenting and killing of dogs’
Reply: we actually had such a statement in an early draft, we decided to remove it as it was such an open door. However, we can include once more; see adapted text

19) l319: Space typo in the first word of the line
Reply: corrected

20) l327: Required by WHO also I presume. Please include if yes.
Reply: the latest WHO technical report series refers to WOAH for vaccine requirements. Hence, we prefer to refer to only one source when it comes to the tripartite (WHO/WOAH/FAO)

21) l330-331: Please improve sentence. Not comprehensible.
Reply: Hopefully, the adapted sentence makes ‘more sense’

22) l332: Does ‘same’ bait mean ‘intestine bait’?
Reply: yes, sentence has been adapted to circumvent misunderstandings

23) l333: Which bait was found to be extremely attractive to dogs?
Reply: we clarified the bait and sachet in this sentence to prevent misunderstandings

24) l346: Include other agencies that specify requirements and are now met.
Reply: we wish we could but actually WOAH is the only agency that has listed minimal safety and efficacy requirements for oral rabies vaccines intended for (free-roaming) dogs. The monograph in Pharm Eu is restricted to red foxes and raccoon dogs. As mentioned before, WHO has basically handed over there ‘responsibilities’ concerning detailed requirements for such animal vaccines to WOAH. Thus, it does not make sense to refer to older WHO documents that are no longer ‘valid’.

25) l349: Include ‘tools used to reach better vaccination coverage efficiently and goal of’
Reply: included

26) l350: Remove the word ‘ambiguous’
Reply: In line with another reviewer, we changed this to ‘ambitious’

Reviewer 2 Report

The article "Efficacy of oral rabies vaccine baits containing SPBN GASGAS in domestic dogs according to international standards" reports definitive experimental demonstration of the efficacy of a live recombinant rabies vaccine delivered to the main vector of rabies, the dog, by the oral route. The article is well written and the data presented clearly.  Could the authors give more clarification in section 2.1 of the composition of the bait (eggs plus what?) and the material of the sachet used, what is "soft blister"? The Discussion highlights the disadvantage of using an aluminium sachet but it is not clear what material has been used.

Minor text edits

2.8.1. Immune Sorbent is usually one word.

4. Discussion, paragraph 3, introduce gap between "vaccine" and "in"

5. Conclusions, do the authors mean "ambiguous" or " ambitious"?

Author Response

Please see below, our reply to the comments made by the reviewer:

Reviewer 2

The article "Efficacy of oral rabies vaccine baits containing SPBN GASGAS in domestic dogs according to international standards" reports definitive experimental demonstration of the efficacy of a live recombinant rabies vaccine delivered to the main vector of rabies, the dog, by the oral route. The article is well written and the data presented clearly. 
Reply: no action necessary

Could the authors give more clarification in section 2.1 of the composition of the bait (eggs plus what?) and the material of the sachet used, what is "soft blister"?
Reply: as the bait and sachet used are proprietary data, we do not want to go into too much details but we added some information so that the readers can understand the term ‘soft blister’ better

The Discussion highlights the disadvantage of using an aluminium sachet but it is not clear what material has been used.
Reply: in the discussion, it is specifically mentioned that it concerns a PVC sachet with an aluminium cover foil. It seems that this is a detailed description. We added the word ‘rigid’ indicating that the shape of the sachet was not flexible like the ‘soft blister’ 

Minor text edits

2.8.1. Immune Sorbent is usually one word.
Reply: corrected

  1. Discussion, paragraph 3, introduce gap between "vaccine" and "in"
    Reply: corrected
  2. Conclusions, do the authors mean "ambiguous" or " ambitious"?
    Reply: actually both, but let us remain diplomatic and change it to ‘ambitious’

Reviewer 3 Report

The title as well as keywords accurately reflects the major findings of the work.

The abstract adequately summarize methodology, results, and significance of the study.

The introduction section is well written and it falls within the topic of the study. Punctuation should be improved.

Material and methods are well written and meticulously describe methodology applied in the study.

Authors should indicate the type of tubes used for blood collection including Manufacture also.

Moreover, Authors should indicate the centrifugation characteristics applied to obtain serum samples and the time between blood sampling and laboratory analysis.

Did Authors store the samples in refrigerator/freezer before analysis? If yes, how long were the samples frozen? and at what temperature? Please clarify these aspects.

Results are well written and well discussed in Discussion section. Appropriate bibliographic information are reported.

The conclusion section is well written, Authors well summarize the main findings gathered in the study, well emphasize the significance of the study, and clearly propose new insights in the investigated field. 

The tables as well as Figures are generally good, well represents the main findings of the study.

Author Response

See below for our point-to-point reply to the reviewers' comments

The title as well as keywords accurately reflects the major findings of the work.
Reply: no action necessary

The abstract adequately summarize methodology, results, and significance of the study.’
Reply: no action necessary

The introduction section is well written and it falls within the topic of the study. Punctuation should be improved.
Reply: we double-checked the punctuation; it seems that the software we used had a different ‘grammar teacher’ than the reviewer as it did not accept any changes. Hence, we decided to change the punctuation in one sentence only (line 58)

Material and methods are well written and meticulously describe methodology applied in the study.
Reply: no action necessary

Authors should indicate the type of tubes used for blood collection including Manufacture also.
Reply: information added

Moreover, Authors should indicate the centrifugation characteristics applied to obtain serum samples and the time between blood sampling and laboratory analysis.
Reply: details of centrifugation added, because samples were stored for different periods, a general statement on duration of storage is included.

Did Authors store the samples in refrigerator/freezer before analysis? If yes, how long were the samples frozen? and at what temperature? Please clarify these aspects.
Reply: statement included.

Results are well written and well discussed in Discussion section. Appropriate bibliographic information are reported.
Reply: no action necessary

The conclusion section is well written, Authors well summarize the main findings gathered in the study, well emphasize the significance of the study, and clearly propose new insights in the investigated field. 
Reply: no action necessary

The tables as well as Figures are generally good, well represents the main findings of the study.
Reply: no action necessary